# Scaling-up a pharmacist-led information technology intervention (PINCER) to reduce hazardous prescribing in general practices: Multiple interrupted time series study

Sarah Rodgers[1,2‡], Amelia C. Taylor[1‡*], Stephen A. Roberts[3], Thomas Allen[4,5], Darren M. Ashcroft[6,7], James Barrett[2], Matthew J. Boyd[8], Rachel A. Elliott[4], Kamlesh Khunti[9], Aziz Sheikh[10], Despina Laparidou[11], Aloysius Niroshan Siriwardena[11], Anthony J. Avery[1,7]

1 School of Medicine, University of Nottingham, Nottingham, United Kingdom, 2 PRIMIS, University of Nottingham, Nottingham, United Kingdom, 3 Centre for Biostatistics, University of Manchester, Manchester, United Kingdom, 4 Manchester Centre for Health Economics, University of Manchester, Manchester, United Kingdom, 5 Danish Centre for Health Economics, Department of Public Health, University of Southern Denmark, Odense, Denmark, 6 Centre for Pharmacoepidemiology and Drug Safety, School of Health Sciences, University of Manchester, Manchester, United Kingdom, 7 NIHR Greater Manchester Patient Safety Translational Research Centre, Manchester, United Kingdom, 8 School of Pharmacy, University of Nottingham, Nottingham, United Kingdom, 9 Diabetes Research Centre, University of Leicester, Leicester, United Kingdom, 10 Usher Institute, University of Edinburgh, Edinburgh, United Kingdom, 11 Community and Health Research Unit, University of Lincoln, Lincoln, United Kingdom

‡ These authors share first authorship on this work.
* amelia.taylor@nottingham.ac.uk

## Abstract

### Background

We previously reported on a randomised trial demonstrating the effectiveness and cost-effectiveness of a pharmacist-led information technology intervention (PINCER). We sought to investigate whether PINCER was effective in reducing hazardous prescribing when rolled out at scale in UK general practices.

### Methods and findings

We used a multiple interrupted time series design whereby successive groups of general practices received the PINCER intervention between September 2015 and April 2017. We used 11 prescribing safety indicators to identify potentially hazardous prescribing and collected data over a maximum of 16 quarterly time periods. The primary outcome was a composite of all the indicators; a composite for indicators associated with gastrointestinal (GI) bleeding was also reported, along with 11 individual indicators of hazardous prescribing. Data were analysed using logistic mixed models for the quarterly event numbers with the appropriate denominator, and calendar time included as a covariate.

PINCER was implemented in 370 (94.1%) of 393 general practices covering a population of almost 3 million patients in the East Midlands region of England; data were successfully extracted from 343 (92.7%) of these practices. For the primary composite outcome, the

**Data Availability Statement:** We are unable to share the practice level data as the data sharing agreement, provided to the practices involved, restricted the use of the data to PRIMIS, the Clinical Commissioning Group and members of the study team. It stated that data would not be shared

with any person or organisation that does not appear in this list. For enquiries about the data, please contact the University of Nottingham via the following email address: 'sponsor@nottingham.ac. uk'.

**Funding:** This study was funded by The Health Foundation (www.health.org.uk, Award Number: 7419) AJA, DMA, MJB, RAE, KK, SR, AS and ANS, and the East Midlands Academic Health Science Network (www.emahsn.org.uk, Award Number: 39701) AJA, SR. DMA was supported by the National Institute for Health and Care Research (NIHR) Greater Manchester Patient Safety Translational Research Centre (award number: PSTRC-2016-003, http://www.patientsafety. manchester.ac.uk). KK is supported by the National Institute for Health and Care Research (NIHR) Applied Research Collaboration East Midlands (https://arc-em.nihr.ac.uk) and the NIHR Leicester Biomedical Research Centre (https://www. leicesterbrc.nihr.ac.uk). The funders had no role in study design, data collection and analysis, decision to publish, or preparation of the manuscript.

**Competing interests:** I have read the journal's policy and the authors of this manuscript have the following competing interests: AJA is the National Clinical Director for Prescribing for NHS England. AJA and RAE hold a NIHR Programme Grants for Applied Research: Avoiding patient harm through the application of prescribing safety indicators in English general practices (acronym: PRoTeCT): RP-PG-1214-10005.MJB is a reviewer for NIHR and Pharmacy Research UK grant awards and is an Editorial Board member for Research in Social and Administrative Pharmacy (RSAP) and Exploratory Research in Clinical and Social Pharmacy (ERCSP). AS is on the editorial board of PLOS Medicine. ANS received funding from the National Institute for Health Research for several unrelated studies listed in publication.

**Abbreviations:** AHSN, Academic Health Science Network; aOR, adjusted odds ratio; CCG, Clinical Commissioning Group; CI, confidence interval; CQC, Care Quality Commission; DOAC, direct oral anticoagulant; eGFR, estimated glomerular filtration rate; GMS, General Medical Services; GI, gastrointestinal; GP, general practitioner; MIQUEST, Morbidity Information and Query Export Syntax; NHS, National Health Service; NICE, National Institute for Health and Care Excellence; NSAID, nonsteroidal anti-inflammatory drug; OR, odds ratio; PCN, Primary Care Network; PINCER, pharmacist-led information technology intervention; PPI, proton pump inhibitor; SD, standard deviation; SMASH, Safety Medication

PINCER intervention was associated with a decrease in the rate of hazardous prescribing of 16.7% (adjusted odds ratio (aOR) 0.83, 95% confidence interval (CI) 0.80 to 0.86) at 6 months and 15.3% (aOR 0.85, 95% CI 0.80 to 0.90) at 12 months postintervention. The unadjusted rate of hazardous prescribing reduced from 26.4% (22,503 patients in the numerator/853,631 patients in the denominator) to 20.1% (11,901 patients in the numerator/ 591,364 patients in the denominator) at 6 months and 19.1% (3,868 patients in the numerator/201,992 patients in the denominator). The greatest reduction in hazardous prescribing associated with the intervention was observed for the indicators associated with GI bleeding; for the GI composite indicator, there was a decrease of 23.9% at both 6 months (aOR 0.76, 95% CI 0.73 to 0.80) and 12 months (aOR 0.76, 95% CI 0.70 to 0.82) postintervention. The unadjusted rate of hazardous prescribing reduced from 31.4 (16,185 patients in the numerator/515,879 patients in the denominator) to 21.2% (7,607 patients in the numerator/ 358,349 patients in the denominator) at 6 months and 19.5% (2,369 patients in the numerator/121,534 patients in the denominator). We adjusted for calendar time and practice, but since this was an observational study, the findings may have been influenced by unknown confounding factors or behavioural changes unrelated to the PINCER intervention. Data were also not collected for all practices at 6 months and 12 months postintervention.

## Conclusions

The PINCER intervention, when rolled out at scale in routine clinical practice, was associated with a reduction in hazardous prescribing by 17% and 15% at 6 and 12 months postintervention. The greatest reductions in hazardous prescribing were for indicators associated with risk of GI bleeding. These findings support the wider national rollout of PINCER in England.

## Author summary

### Why was this study done?

- Prescribing errors in general practice are a preventable cause of patient morbidity, hospitalisations, and deaths.

- The pharmacist-led information technology intervention (PINCER) aims to reduce hazardous prescribing by searching general practice clinical computer systems to identify patients at risk from hazardous prescribing and pharmacists working with the practices to ameliorate these.

- In a large cluster randomised trial, the PINCER intervention was found to be an effective and cost-effective method for reducing hazardous prescribing in general practice.

- The current study was done to evaluate whether the intervention would be effective when rolled out at scale.

Dashboard; TPP, The Phoenix Partnership; UK, United Kingdom; WHO, World Health Organization.

### What did the researchers do and find?

- PINCER was implemented in 370 (94.1%) of 393 general practices in the East Midlands of England between September 2015 and April 2017; data were successfully extracted from 343 (92.7%) of these practices.

- We used 11 prescribing safety indicators to identify potentially hazardous prescribing and collected data over a maximum of 16 quarterly time periods.

- We used a multiple interrupted time series design, where the rate of hazardous prescribing before the intervention was compared to 6 months and 12 months postintervention. The data was adjusted for calendar time and general practice.

- The PINCER intervention was associated with an overall decrease in the rate of hazardous prescribing of 16.7% at 6 months and 15.3% at 12 months postintervention.

- We were not able to collect 6- and 12-month follow-up data on all practices, and this is the main limitation of the study.

### What do these findings mean?

- The findings suggest that the PINCER intervention was effective when rolled out at scale.

- These findings support the wider national rollout of PINCER in England and may help to inform policy makers when considering implementation of similar interventions.

## Introduction

Medication errors in general practice are an important and expensive preventable cause of patient safety incidents associated with morbidity, hospitalisations, and deaths. A study in English general practices identified errors in 5% of prescription items, with one in 550 items containing a potentially life-threatening error [1]. Further studies have shown hazardous prescribing in general practices to be a contributory cause of around one in 25 hospital admissions [2]. Preventable adverse drug events leading to a hospital admission are estimated to cost £83.7 million and to cause 627 deaths in England each year [3].

The World Health Organization (WHO) has identified "Medication Without Harm" as the theme for their Third Global Patient Safety Challenge, which aims to reduce severe avoidable medication-related harm by 50% globally by targeting healthcare providers' behaviour, systems and practices of medication, medicines, and the public [4]. In response to this challenge, the Department of Health and Social Care in England commissioned a report on the prevalence and cost of medication errors, which estimated that 66 million potentially clinically significant errors occur per year, 71% of which are in primary care [3]. There is therefore a need to develop and implement interventions to reduce medication error associated with avoidable harm.

The PINCER intervention (a pharmacist-led information technology intervention for medication errors in general practice) involves searching general practitioner (GP) clinical systems

using automated computerised hazardous prescribing indicators to identify patients at risk from their prescriptions, and then acting to correct the problems and to minimise future risk with pharmacist support. Our cluster randomised controlled trial found the PINCER intervention to be acceptable, effective, and cost-effective in reducing rates of hazardous prescribing [5]. At six months' follow-up, the general practices receiving computerised feedback and pharmacist support had significantly less hazardous prescribing than those that received computerised feedback alone.

Since the original PINCER study, work has been undertaken to further refine the intervention and to implement the intervention at scale across the East Midlands region of England. The hazardous prescribing indicators were updated, based on a systematic review that identified 12 drug groups accounting for 80% of medication-related and preventable hospital admissions [2]. Three drug groups, anticoagulants, antiplatelets, and nonsteroidal anti-inflammatory drugs (NSAIDs) (which all cause gastrointestinal (GI) bleeding), were found to be responsible for over one-third of these admissions. An important implication from this review is that reducing hazardous prescribing in general practice associated with specific groups of drugs could prevent most medication-related hospital admissions.

Therefore, we developed a set of prescribing safety indicators [6] to identify patients exposed to medication errors in general practice, and the PINCER intervention is designed to ameliorate risk from the most common and important of these errors. We sought to assess whether PINCER was effective at reducing rates of hazardous prescribing when rolled out at scale across the East Midlands in the real-world setting. Our hypothesis was that the intervention would result in clinically important, sustained reductions in such hazardous prescribing.

## Methods

### Study design and participants

We used a multiple interrupted time series design whereby successive groups of general practices received the PINCER intervention.

Across the East Midlands of England, 12 Clinical Commissioning Groups (CCGs), which are National Health Service (NHS) bodies responsible for the commissioning of healthcare services in local areas, agreed to participate. All general practices in these CCGs were invited to take part in the knowledge that they used electronic health records with embedded electronic prescribing capability. Eleven general practices were not eligible for inclusion in the analysis as they had previously piloted the hazardous prescribing indicators to be used in the study.

### The intervention

The PINCER intervention [5] comprises three components. First, the computer systems of general practices are searched to identify patients at risk of potentially hazardous prescribing using a set of prescribing safety indicators (Table 1). Second, pharmacists, specifically trained to deliver the intervention, provide an educational outreach intervention where they meet with GPs and other practice staff to:

• Discuss the search results and highlight the importance of the hazardous prescribing identified using brief educational materials. These feedback sessions were to be held straight after running the searches and then at regular intervals. The educational materials included training on root cause analysis using well-established techniques such as the fishbone diagram and the 5 whys.

**Table 1. The prescribing indicators grouped by serious harm outcome.**

| Indicator | Description of Indicator | Group at risk (Denominator) | Group exposed to hazardous prescribing (Numerator) |
|---|---|---|---|
| **Hazardous prescribing indicators associated with GI bleeding** | | | |
| A | Prescription of an oral NSAID, without coprescription of an ulcer healing drug, to a patient aged ≥65 years | Patients aged ≥65 years without coprescription of an ulcer-healing drug (PPI or $H_2$ antagonist) in the 3 months leading up to the audit date | Patients prescribed an oral NSAID in the 3 months leading up to the audit date |
| B | Prescription of an oral NSAID, without coprescription of an ulcer-healing drug, to a patient with a history of peptic ulceration | Patients aged ≥18 years with a Read code for peptic ulcer or upper GI bleed at least 3 months before audit date and not prescribed an ulcer-healing drug (PPI or $H_2$ antagonist) within the 3 months leading up to the audit date | Patients prescribed an oral NSAID within the 3 months leading up to the audit date |
| C | Prescription of an antiplatelet drug, without coprescription of an ulcer-healing drug, to a patient with a history of peptic ulceration | Patients aged ≥18 years with a Read code for peptic ulcer or GI bleed at least 3 months before audit date and not prescribed an ulcer-healing drug (PPI or $H_2$ antagonist) within the 3 months leading up to the audit date | Patients prescribed an antiplatelet drug (aspirin or clopidogrel or prasugrel or ticagrelor) within the 3 months leading up to the audit date |
| D | Prescription of warfarin or DOAC in combination with an oral NSAID | Patients aged ≥18 years prescribed warfarin or a DOAC (apixaban or dabigatran or rivaroxaban) within the 3 months leading up to the audit date | Patients prescribed an oral NSAID within the 3 months leading up to the audit date |
| E | Prescription of warfarin or DOAC and an antiplatelet drug in combination without coprescription of an ulcer-healing drug | Patients aged ≥18 years prescribed warfarin or DOAC without coprescription of an ulcer-healing drug (PPI or $H_2$ antagonist) within the 3 months leading up to the audit date | Patients prescribed an antiplatelet drug (aspirin or clopidogrel or prasugrel or ticagrelor) within the 3 months leading up to the audit date and within 28 days of the warfarin/DOAC prescription |
| F | Prescription of aspirin in combination with another antiplatelet drug (without coprescription of an ulcer-healing drug) | Patients aged ≥18 years prescribed aspirin without coprescription of an ulcer-healing drug (PPI or $H_2$ antagonist) within the 3 months leading up to the audit date | Patients prescribed another antiplatelet drug (clopidogrel or prasugrel or ticagrelor) within the 3 months leading up to the audit date and within 28 days of the aspirin prescription |
| **Hazardous prescribing indicators associated with asthma** | | | |
| G | Prescription of a nonselective β-blocker to a patient with asthma | Patients aged ≥18 years with a Read code for asthma at least 3 months before audit date and no subsequent asthma resolved code during that time period | Patients prescribed a nonselective β-blocker within the 3 months leading up to the audit date |
| H | Prescription of a long-acting beta-2 agonist inhaler (excluding combination products with inhaled corticosteroid) to a patient with asthma who is not also prescribed an inhaled corticosteroid | Patients aged ≥18 years with a Read code for asthma at least 3 months before audit date (and no subsequent asthma resolved code during that time period) who have been prescribed a long-acting beta-2 agonist inhaler (excluding combination products with inhaled corticosteroid) within the last 3 months | Patients not prescribed an inhaled corticosteroid within the 3 months leading up to the audit date |
| **Hazardous prescribing indicators associated with heart failure** | | | |
| I | Prescription of an oral NSAID to a patient with heart failure | Patients aged ≥18 years who have a diagnosis of heart failure at least 3 months before the audit date | Patients prescribed an oral NSAID within the 3 months leading up to the audit date |
| **Hazardous prescribing indicators associated with cardiovascular events, including stroke** | | | |
| J | Prescription of antipsychotics for >6 weeks in a patient aged ≥65 years with dementia but not psychosis | Patients aged ≥65 years with a Read code for dementia at least 3 months before the audit date and no Read code for psychosis (or have a psychosis Read code and a subsequent psychosis resolved Read code) at least 3 months before the audit date | Patients prescribed antipsychotic drugs at least once within the 3 months leading up to the audit date |
| **Hazardous prescribing indicators associated with acute kidney injury** | | | |
| K | Prescription of an oral NSAID to a patient with eGFR <45 mL/min | Patients aged ≥18 years with chronic renal failure: eGFR <45 mL/min at least 3 months before the audit date | Patients prescribed an oral NSAID within the 3 months leading up to the audit date |

*(Continued)*

**Table 1.** (Continued)

| Indicator | Description of Indicator | Group at risk (Denominator) | Group exposed to hazardous prescribing (Numerator) |
|---|---|---|---|
| Composite of all indicators (A-K) | Composite indicator (all 11 indicators) | Number of unique patients in any of the above "at risk" groups (denominators) | Number of patients exposed to a high-risk prescription (sum of the number of patients included in indicators A-K) |
| Composite of GI bleed indicators (A-F) | Composite of GI bleed indicator (all GI bleed indicators) | Number of unique patients in any of the above "at risk" groups (denominators associated with a GI bleed) | Number of patients exposed to a high-risk prescription (sum of the number of patients included in indicators A-F) |

DOAC, direct oral anticoagulant; eGFR, estimated glomerular filtration rate; GI, gastrointestinal; NSAID, nonsteroidal anti-inflammatory drug; PPI, proton pump inhibitor.

- Agree on an action plan, retained within the practice, for reviewing patients identified as high risk and improving prescribing and medication monitoring systems using root cause analysis to minimise future risk.

Third, the pharmacists (sometimes supported by pharmacy technicians) work with, and support, general practice staff to implement the agreed action plan, sometimes making the necessary changes themselves.

The East Midlands rollout was a pragmatic implementation study, and the PINCER intervention was delivered by the pharmacists employed in each of the CCG Medicines Optimisation teams. Training on the PINCER intervention and CHART software was delivered to members of the Medicines Optimisation Teams in all 12 CCGs that implemented the PINCER intervention. The time that the pharmacists spent varied by CCG, depending on the resourcing level of each of the CCG Medicines Optimisation Team. For example, we know some localities were very well resourced in terms of numbers of CCG pharmacists, whereas other localities had very little CCG pharmacist resource.

Since the original PINCER study [5], further changes were made to scale the intervention including updating training materials. PRIMIS at the University of Nottingham wrote computerised queries using Morbidity Information and Query Export Syntax (MIQUEST) software to identify patients at risk of hazardous prescribing for each of the prescribing safety indicators. The pharmacists working in the general practices received training on how to run the MIQUEST queries using the general practice clinical information system and send aggregate summative data to PRIMIS via secure transfer for inclusion in a comparative analysis service (CHART and CHART Online software; [7]). To evaluate the intervention, quarterly retrospective data were collected towards the end of the study using a modified set of MIQUEST queries. Retrospective data collection was undertaken either by the pharmacists working in the general practices or by PRIMIS (who, with permission from the CCGs and practices, remotely extracted the data). Data extracted from the general practices were transferred into an electronic database and were processed (using similar approaches to those that we adopted in the PINCER trial; [5]) in readiness for analysis. This included collation of numerators and denominators for each of the indicators for each of the time points, and calculation of composite indicators.

## Outcome measures

We prespecified 11 prescribing safety indicators to identify potentially hazardous prescribing, described in Table 1. These 11 indicators are associated with the following serious harm outcomes: GI bleeding, asthma, heart failure, stroke, and acute kidney injury. The number of

patients in each practice exposed to each of the 11 indicators and the number of patients in groups classified as being at risk of being exposed were collected. Data were collected retrospectively at quarterly time points over the period of 4 years (30 November 2013 and 31 August 2017). However, not all practices contributed data at all 16 time points. The implementation quarter for each practice are presented in the Supporting information (S1 Appendix). For each practice, the implementation quarter was assigned Quarter 0, postimplementation quarters were assigned Q1, Q2, Q3, etc., and preimplementation quarters were assigned Q-1, Q-2, Q-3, etc.

This yielded numerator and denominator data for each practice for each of the prescribing safety indicators for each quarter. The numerator and denominator data were used to calculate the proportion of patients at risk who were exposed to hazardous prescribing for each indicator in each GP practice over a maximum of 16 quarters (4 years). Composite measures were defined for patients exposed to any of the indicators and for the combined GI indicators (Table 1). For the numerator, the sum of the number for patients exposed to each type of hazardous prescribing included in the composite was used, and for the denominator, the number of patients at risk of exposure to the hazardous prescribing indicators was used. This allowed a proportion of patients exposed to be calculated. The primary outcome was the composite of all the indicators. Secondary outcomes were the composite of the indicators associated with GI bleeds, and each of the 11 indicators. The GI indicators were singled out as a group because there was a relatively large number of these, and other studies have suggested these may be particularly sensitive to intervention (reference DQIP [7] and SMASH [8]).

## Statistical methods

Although we used a multiple interrupted time series design, we did not conduct a classic interrupted time series analysis. Instead, the model we used includes a linear temporal effect over the preintervention period and estimates a preintervention effect from this; the postintervention effects are the differences from this extrapolated trend. We were not expecting a sustained effect from the intervention, so we did not use the classic linear postintervention effect (step plus change in slope) but rather parameterised the model to allow for a nonlinear intervention effect.

Analyses were conducted on an intention-to-treat basis, with the assumption that practices became exposed to the intervention at the time the intervention was introduced (defined as the time when the first computer search was conducted in each practice to identify patients at risk). The assumption was also made that the intervention start date was the date that practices uploaded their data to CHART Online as part of the rollout of the intervention. This date was then used to indicate the implementation date for the retrospective data collection. Practices were included in the analysis irrespective of the degree to which they engaged in the PINCER intervention.

Where there were quarters at the extremes of the data collection period with very few practices (<10), a pragmatic decision was taken prior to any formal analysis to exclude these from the analysis. The number of practices included in each quarter aligned by implementation quarter are presented in the Supporting information (S1 Appendix).

Prior to analysis, outcome data were summarised graphically by calendar quarter and by time since the intervention, presented as a mean rate across all practices.

Formal analysis of hazardous prescribing for each of the single and the composite indicators utilised a mixed model approach, with logistic mixed models for the quarterly event numbers with the appropriate denominator. As the intervention was expected to be delivered over the course of a few weeks and the effects of this to diminish over time [5], a priori we wished to

allow for a time-dependent intervention effect, rather than the usual segmented linear assumptions. Therefore, we represented the time postintervention as a categorical variable coding each postintervention quarter; all preintervention quarters were assigned a single reference level for the treatment effect (noting that the fitted outcomes will still vary according to the fitted secular trend). Preimplementation base rates and numbers at risk are presented as the mean of the four quarters prior to the implementation at each site. The quarter during which the intervention began was treated as the first postintervention quarter. The intervention will have occurred at some point during this quarter. Calendar time (to account for secular trends) was included as a covariate, modelled as a linear function of the calendar quarter. Random intercept terms for GP practice allowed for within practice correlations. Estimation of confidence intervals (CIs) utilised the robust standard errors approach.

Specific hypotheses of improvements over temporal trends at 6 months (primary) and 12 months (secondary) corresponded to the assessment times in the previous cluster randomised trial [5]. These were tested by constructing appropriate contrasts comparing the second and fourth quarters, where the quarter at zero is the quarter during which the implementation was introduced, with the preimplementation level. Effect sizes are presented as (adjusted) odds ratios (aORs) compared with the preintervention period. The model is described explicitly in the Supporting information (S2 Appendix).

Data were analysed using the statistical package Stata SE 16.

This study is reported as per the Strengthening the Reporting of Observational Studies in Epidemiology (STROBE) guideline (S3 Appendix).

## Ethical approval

The study received approval from the University of Nottingham Research Ethics Committee on 23 January 2017 (reference number A16012017).

## Patient and public involvement

In developing our funding application for this project, we took account of the views of patients and members of the public, and the East Midlands Patient and Public Involvement Senate was a partner on the application. Three Patient and Public Involvement representatives were involved throughout the project and actively contributed to ongoing discussions about the conduct of the study. The Patient and Public Involvement representatives attended the quarterly Project Steering Committee meetings and Evaluation Advisory Committee meetings to provide oversight for the project. They supported and checked project information sheets prior to submission for ethical review and contributed to the development of the interview and focus group guides used in the qualitative study and the interpretation of the findings.

## Results

Of the 393 general practices in the 12 participating CCGs in the East Midlands (October 2015), PINCER was implemented in 370 (94.1%) between September 2015 and April 2017. Ten practices had closed by the end of the implementation period, and 11 practices from one CCG were involved in piloting the PINCER indicators and were excluded from analysis. Therefore, 349 GP practices were eligible for retrospective data collection of which data were collected for 343 (92.7% of those that implemented PINCER and 98.0% of those eligible for data collection). To be included, a practice must have run the intervention and uploaded at least one quarter of data. Therefore, not all practices contributed data at all 16 time points. We had data for all 343 practices for between three and seven quarters before the start of the intervention, but due to some practices uploading their own data to CHART Online, there were some gaps in the data

in the postintervention quarters, including quarter 0. The reduction in the number of practices at follow-up was not believed to be related to the practices' engagement with PINCER, but instead whether they uploaded the data to CHART Online, to be shared with the research team. The duration of the postintervention data collected for each practice was limited for those practices that implemented the intervention towards the end of the study period and where practices stopped uploading data during the follow-up period. No postimplementation data were collected in seven practices. At 6 months follow-up, data for 212 practices were collected, and by 12 months follow-up, this had reduced to 70 (S4 Appendix). The missing data at follow-up does not seem to be related to the practice characteristics (Table 2). We looked at the characteristics of the 212 practices included at 6 months postintervention and the 70 practices included at 12 months postintervention and found these to be comparable with the 343 participating practices. As a sensitivity analysis, we also repeated the analysis including only practices with at least 6 months of follow-up data. A similar effect size was noted at both 6 and 12 months postintervention (S5 Appendix). This was expected as the intervention effect estimates using the entire data set were derived from those practices that had sufficient follow-up. We also compared the raw and adjusted preimplementation rates for those practices with up to 6 months and up to 12 months follow-up (S6 Appendix). The rates for each group of practices were found to be similar.

The rollout of the intervention was phased for each CCG (S1 Appendix). Characteristics of the participating general practices are shown in Table 2. In most respects, these were similar to English general practices as a whole, but study practices had higher mean list size and GP whole time equivalents, were more likely to use The Phoenix Partnership (TPP) SystmOne clinical system, and more likely to be rated "outstanding" by the Care Quality Commission (CQC).

At baseline (the quarter prior to the quarter when the intervention was implemented in each practice), a total of 2.97 million patient records were searched, and 22,105 instances of potentially hazardous prescribing were identified in 21,283 patients. Table 3 shows the number of patients in the numerator and denominator for each indicator at baseline, along with the rate of hazardous prescribing per 1,000 patients at risk.

A decrease in hazardous prescribing rates over time was observed across the participating practices for most indicators (Fig 1) for the overall composite outcome measure and the GI composite outcome measure.

Table 4 shows the number of patients at risk preimplementation and the rate of potentially hazardous prescribing preintervention and at 6 and 12 months postimplementation. Odds ratios, adjusted for GP practice and calendar time, are given for differences in rates of hazardous prescribing at 6 and 12 months compared to preintervention. There was a reduction in hazardous prescribing for the composite outcomes and most of the individual outcomes at both 6 and 12 months postintervention. For the primary composite outcome, the PINCER intervention was associated with a decrease in the rate of hazardous prescribing of 16.7% (aOR 0.83, 95% CI 0.80 to 0.86) at 6 months and 15.3% (aOR 0.85, 95% CI 0.80 to 0.90) at 12 months postintervention. The unadjusted rate of hazardous prescribing reduced from 26.4 (22,503 patients in the numerator/853,631 patients in the denominator) to 20.1% (11,901 patients in the numerator/591,364 patients in the denominator) at 6 months and 19.1% (3,868 patients in the numerator/201,992 patients in the denominator).

For the GI composite indicator, there was a decrease of 23.9% at both 6 months (aOR 0.76, 95% CI 0.73 to 0.80) and 12 months (aOR 0.76, 95% CI 0.70 to 0.82) postintervention. The unadjusted rate of hazardous prescribing reduced from 31.4 (16,185 patients in the numerator/515,879 patients in the denominator) to 21.2% (7,607 patients in the numerator/358,349 patients in the denominator) at 6 months and 19.5% (2,369 patients in the numerator/121,534 patients in the denominator).

**Table 2. Characteristics of 343 general practices in the East Midlands that implemented PINCER and were included in the analysis compared with English general practices overall.**

| Characteristics | | Study practices (*n* = 343) | Study practices included at 6 months (*n* = 212) | Study practices included at 12 months (*n* = 70) | All English practices |
|---|---|---|---|---|---|
| List size: Mean (SD) | | 8,096 (4,777.9) | 8,516 (5,018.8) | 9,205 (6,477.1) | 7,586[a] |
| Clinical computer system: Number (%) | TPP | 263 (76.7) | 152 (71.6) | 54 (77.1) | (30)[b] |
| | EMIS | 80 (23.3) | 60 (28.3) | 16 (22.9) | (56)[b] |
| Index of multiple deprivation: Mean (SD)[c] | | 22.9 (11.1) | 19.7 (10.0) | 22.9 (11.0) | 21.8 |
| Percentage of patients from minority ethnic groups: Mean (SD)[d] | | 16.6 (21.3) | 9.8 (13.3) | 10.9 (14.1) | 14.0 |
| GP whole time equivalent (per 1,000 patients): Mean (SD)[e] | | 4.65 (3.10) | 4.83 (3.12) | 5.3 (3.9) | 4.16 |
| Quality and Outcomes Framework score 2016/2017: Mean (SD) out of 559 points available[f] | | 537 (29.0) | 539 (26.7) | 536 (28.6) | 534 |
| CQC Safety Rating: percentage[g] | Outstanding | 12 (40 practices) | 12 (26 practices) | 9 (6 practices) | 1 |
| | Good | 78 (268 practices) | 78 (165 practices) | 71 (50 practices) | 84 |
| | Requires improvement | 7 (23 practices) | 7 (14 practices) | 14 (10 practices) | 13 |
| | Inadequate | 1 (4 practices) | 2 (4 practices) | 3 (2 practices) | 2 |

CQC, Care Quality Commission; EMIS, Egton Medical Information Supplies; PINCER, pharmacist-led information technology intervention; SD, standard deviation; TPP, The Phoenix Partnership.

[a]List size accessed from Public Health England National General Practice Profiles for 2016. http://fingertips.phe.org.uk/profile/general-practice/data

[b]Estimated share of primary care (general practice) clinical computer system market 2017. https://www.emisgroupplc.com/media/1420/emis-group-plc-final-year-results-2017-presentation.pdf

[c]Index of Multiple Deprivation for 2015. Accessed from Public Health England National General Practice Profiles. http://fingertips.phe.org.uk/profile/general-practice/data

[d]Ethnicity: Taken from 2011 Census. Accessed from Public Health England National General Practice Profiles. http://fingertips.phe.org.uk/profile/general-practice/data

[e]Whole Time Equivalents: Taken from NHS Digital March 2017. https://digital.nhs.uk/data-and-information/publications/statistical/general-practice-workforce-archive/high-level-march-2017-provisional-experimental-statistics

[f]Quality and Outcomes Framework score 2016/2017: Accessed from Public Health England National General Practice Profiles. http://fingertips.phe.org.uk/profile/general-practice/data

[g]Care Quality Commission Safety Rating May 2017; Taken from Care Quality Commission: http://www.cqc.org.uk/sites/default/files/20170921_state_of_care_in_general_practice2014-17.pdf

Fig 2 shows the ORs for the overall composite outcome and the GI composite outcome over time, with a drop over the first two quarters after implementation, which was sustained over the whole study period. The plots for the individual indicators are included in S7 Appendix. The 95% CIs are much larger postintervention than preintervention. This is due to the number of preintervention quarters that were combined to contribute to the preintervention data point, where only single quarters contributed to each of the postimplementation data points.

No reduction in the rate of hazardous prescribing was found for indicators associated with asthma (G and H) or stroke (J).

## Discussion

The PINCER intervention, when rolled out and evaluated at scale to 343 general practices, was associated with reductions of hazardous prescribing for the overall composite indicator of 17% and 15% at 6 and 12 months postintervention, respectively, and 24% for the GI bleed composite indicator at both 6 and 12 months postintervention. The observed reductions in hazardous prescribing for the composite indicators were sustained up until 12 months postintervention.

**Table 3. Number of "at-risk" patients identified at baseline (*n* = 370 practices).**

| Prescribing safety indicator | | At risk of hazardous prescribing (denominator) | Exposed to hazardous prescribing (numerator) | Rate of hazardous prescribing per 1,000 patients at risk |
|---|---|---|---|---|
| Outcome: GI bleed | | | | |
| A | Patients aged ≥65 years prescribed an oral NSAID without coprescription of an ulcer-healing drug | 369,090 | 8,281 | 22 |
| B | Patients aged ≥18 years with a history of peptic ulceration prescribed an oral NSAID without coprescription of an ulcer-healing drug | 20,668 | 471 | 23 |
| C | Patients aged ≥18 years with a history of peptic ulceration prescribed an antiplatelet drug without coprescription of an ulcer-healing drug | 20,668 | 1,638 | 79 |
| D | Patients aged ≥18 years prescribed warfarin or DOAC in combination with an oral NSAID | 51,551 | 635 | 12 |
| E | Patients aged ≥18 years prescribed warfarin or DOAC and an antiplatelet drug in combination without coprescription of an ulcer healing drug | 32,829 | 1,222 | 37 |
| F | Patients aged ≥18 years prescribed aspirin in combination with another antiplatelet drug without coprescription of an ulcer-healing drug | 65,288 | 3,009 | 46 |
| Outcome: Exacerbation of asthma | | | | |
| G | Patients aged ≥18 years with a Read code for asthma prescribed a nonselective beta-blocker | 284,523 | 2,692 | 9 |
| H | Patients aged ≥18 years with a Read code for asthma prescribed a long-acting beta-2 agonist inhaler but not also prescribed an inhaled corticosteroid | 5,217 | 868 | 166 |
| Outcome: Heart failure | | | | |
| I | Patients aged ≥18 years who have a diagnosis of heart failure prescribed an oral NSAID | 23,842 | 467 | 20 |
| Outcome: Stroke | | | | |
| J | Patients aged ≥65 years with a Read code for dementia but no Read code for psychosis prescribed antipsychotic drugs for >6 weeks | 21,931 | 2,094 | 95 |
| Outcome: Kidney Injury | | | | |
| K | Patients aged ≥18 years with an eGFR <45 prescribed an oral NSAID | 35,419 | 728 | 21 |
| Composite of all indicators (including all exposures to hazardous prescribing identified by indicators A-K) | | 931,026 | 22,105 | 24 |
| Composite of GI bleed indicators (including all exposures to hazardous prescribing identified by indicators A-F) | | 560,094 | 15,256 | 27 |

DOAC, direct oral anticoagulant; GI, gastrointestinal; NSAID, nonsteroidal anti-inflammatory drug; PPI, proton pump inhibitor.

This is one of the largest studies that has evaluated the effectiveness of widespread implementation of a medication safety intervention in primary care. The greatest reductions were apparent for indicators targeting hazardous prescribing of NSAIDs or antiplatelets in patients with risk factors for GI bleeding who were not also prescribed an ulcer-healing drug (Indicators A, B, C, E, and F). The only other indicators where a statistically significant reduction in hazardous prescribing was observed to be associated with the intervention was for the prescription of an NSAID to a patient with heart failure (Indicator I) at 6 months (but not 12 months) postintervention and for the prescription of an NSAID to a patient with chronic kidney disease (Indicator K) at 12 months postintervention.

In contrast, no reduction in the rate of hazardous prescribing at 6 months postintervention was found for indicators D (NSAID and anticoagulant prescribed concurrently), G and H (asthma patients prescribed nonselective beta-blockers and asthma patients prescribed a long-

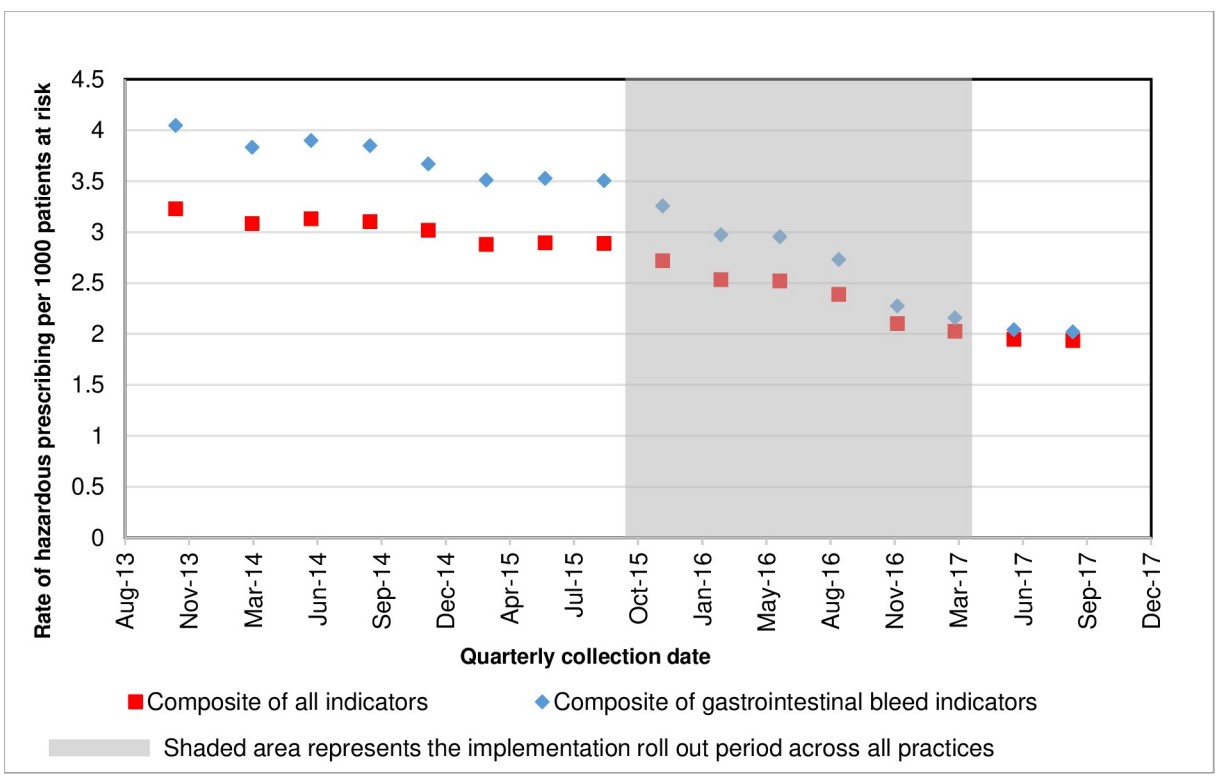

**Fig 1. Hazardous prescribing rate by calendar quarter for the composite prescribing indicators using data from all practices combined.**

acting beta-2 agonist inhaler but not also prescribed an inhaled corticosteroid), and J (dementia patients prescribed antipsychotics). We can only speculate on the reasons for this, with the most likely one being that pharmacists found it more difficult to make these changes.

In our original PINCER trial [5], a more limited set of prescribing safety indicators was used. For the composite of these prescribing indicators there was a reduction in hazardous prescribing of 29% and 22% at 6 and 12 months postintervention, respectively. There are two prescribing indicators that are comparable between the two studies. In the PINCER trial, there was a reduction in the prescribing of NSAIDs to patients with a history of peptic ulcer and no coprescription of a proton pump inhibitor (PPI) (indicator B in the current study) of 42% at 6 months postintervention and a nonstatistically significant 9% at 12 months. Also, in the PINCER trial, there was a reduction in the prescription of beta-blockers in patients with asthma (similar to Indicator G in the current study) of 27% and 22% at 6 and 12 months postintervention, respectively. Comparing the two studies, there was a greater reduction in hazardous prescribing of NSAIDs in patients with a history of peptic ulcer in the PINCER trial at 6 months postintervention, but, unlike the current study, this was not sustained at 12 months postintervention. In the PINCER trial, there was a sustained reduction in the prescription of beta-blockers in patients with asthma postintervention, but this was not seen in the current study.

The DQIP trial used a set of prescribing indicators with some similarities to those used in the current study [8]. In this study involving 33 general practices, there was a 37% reduction in hazardous prescribing and reductions were sustained over the 12-month intervention period. In common with our current study, the greatest reductions were observed for six of the nine indicators where an ulcer-healing drug had not been prescribed when there was a medication-related increased risk of GI bleeding. In addition, the DQIP intervention was effective at reducing the prescription of NSAIDs to patients with chronic kidney disease.

**Table 4. Comparing the rate of hazardous prescribing at 6 months and 12 months postintervention to preintervention, adjusted for general practice and calendar time.**

| Outcome | Preimplementation | | 6 Months | | | 12 Months | | |
|---|---|---|---|---|---|---|---|---|
| | Numerator[a]/ Denominator[a] (Raw Rate[b]) | Fitted rate[b,c] | Numerator[a]/ Denominator[a] (Raw Rate[b]) | Fitted rate[b,c] | OR (95% CI)[d] | Numerator[a]/ Denominator[a] (Raw Rate[b]) | Fitted rate[b,c] | OR (95% CI)[d] |
| Overall composite | 22,503/853,631 (26.4) | 26.7 (26.6:26.7) | 11,901/591,364 (20.1) | 22.3 (22.3:22.4) | 0.83 (0.80:0.86) | 3,868/201,992 (19.1) | 22.7 (22.6:22.8) | 0.85 (0.8: 0.90) |
| GI composite | 16,185/515,879 (31.4) | 32.4 (32.4:32.4) | 7,607/358,349 (21.2) | 24.8 (24.8:24.9) | 0.76 (0.73:0.80) | 2,369/121,534 (19.5) | 24.8 (24.7:24.9) | 0.76 (0.7: 0.82) |
| Indicator A | 8,977/333,980 (26.9) | 26.3 (26.3:26.4) | 3,921/233,533 (16.8) | 19.3 (19.3:19.4) | 0.73 (0.69:0.77) | 1,015/78,187 (13.0) | 18.1 (17.9:18.2) | 0.68 (0.60:0.77) |
| Indicator B | 573/22,484 (25.5) | 23.7 (23.6:23.7) | 274/14,700 (18.6 | 19.0 (18.9:19.1) | 0.80 (0.69:0.92) | 76/5,158 (14.7) | 16.9 (16.7:17.2) | 0.71 (0.57:0.89) |
| Indicator C | 1,947/22,484 (86.6) | 78.9 (78.8:79.0) | 954/14,700 (64.9) | 63.4 (63.3:63.5) | 0.79 (0.73:0.85) | 307/5,158 (59.5) | 59.4 (59.2:59.5) | 0.74 (0.66:0.83) |
| Indicator D | 44,569/44,569 (13.7) | 12.0 (11.9:12.1) | 470/36,014 (13.1) | 11.9 (11.8:12.0) | 0.99 (0.88:1.11) | 180/12,700 (14.2) | 11.2 (11.0:11.4) | 0.93 (0.80:1.09) |
| Indicator E | 11,114/28,507 (39.1) | 38.4 (38.4:38.5) | 578/22,289 (25.9) | 27.0 (26.9:27.1) | 0.70 (0.62:0.77) | 251/7,729 (32.5) | 30.4 (30.2:30.5) | 0.78 (0.66:0.92) |
| Indicator F | 2,962/63,855 (46.4) | 45.4 (45.3:45.4) | 1,410/37,113 (38.0) | 36.7 (36.7:36.8) | 0.80 (0.75:0.86) | 540/12,602 (42.9) | 39.1 (39.0:39.3) | 0.86 (0.76:0.97) |
| Indicator G | 2,311/260,050 (8.9) | 7.6 (7.5:7.6) | 1,750/178,020 (9.8) | 7.3 (7.2:7.4) | 0.96 (0.91:1.02) | 710/62,689 (11.3) | 7.7 (7.6:7.8) | 1.02 (0.94:1.11) |
| Indicator H | 884/5,571 (158.7) | 184.6 (184.5:184.7) | 504/2,961 (170.2) | 194.0 (193.8:194.1) | 1.06 (0.95:1.19) | 120/688 (174.4) | 165.9 (165.7:166.2) | 0.88 (0.70:1.10) |
| Indicator I | 498/22,019 (22.6) | 20.5 (20.4:20.5) | 285/16,120 (17.7) | 18.2 (18.1:18.4) | 0.89 (0.80:0.99) | 91/5,238 (17.4) | 16.8 (16.6:17.1) | 0.82 (0.66:1.02) |
| Indicator J | 1,903/19,837 (95.9) | 81.6 (81.5:81.7) | 1,358/14,476 (93.8) | 86.6 (86.5:86.7) | 1.07 (0.99:1.15) | 461/4,843 (95.2) | 93.8 (93.7:93.9) | 1.16 (1.04:1.30) |
| Indicator K | 722/30,275 (23.9) | 20.6 (20.5:20.7) | 397/21,438 (18.5) | 19.3 (19.1:19.4) | 0.93 (0.83:1.05) | 117/7,000 (16.7) | 17.1 (16.9:17.3) | 0.83 (0.68:1.00) |

[a]Number at risk and preintervention rates estimated as mean over the 4 quarters prior to intervention at each site.

[b]Rate per 1,000 patients at risk.

[c]Fitted rates are adjusted for calendar time and general practice.

[d]Odds ratio relative to preimplementation.

The Safety Medication Dashboard (SMASH) is a pharmacist-led electronic audit and feedback intervention that identifies patients "at risk" of serious adverse events and is based on the PINCER intervention to reduce hazardous prescribing in general practices [9]. In an evaluation undertaken in 43 general practices in Salford, UK using SMASH [10], overall reductions in hazardous prescribing were 28% at 24 weeks postintervention and 41% at 12 months postintervention. For individual indicators, the findings were remarkably consistent with those found in the current study (although reductions were generally greater with SMASH). One exception was that SMASH was associated with reductions in prescribing of NSAIDs to patients with chronic kidney disease.

Overall, these studies confirm that these complex but pragmatic interventions can be effective at reducing hazardous prescribing, but the effects are greatest where a straightforward action can be taken (such as prescribing an ulcer-healing drug to a patient at risk of medication-related GI bleed) compared with stopping (or changing) a medication that may be perceived to be providing benefit to a patient, despite the risk (such as prescribing a beta-blocker in a patient with asthma). Nevertheless, in the current study, SMASH [9] and DQIP [8] studies,

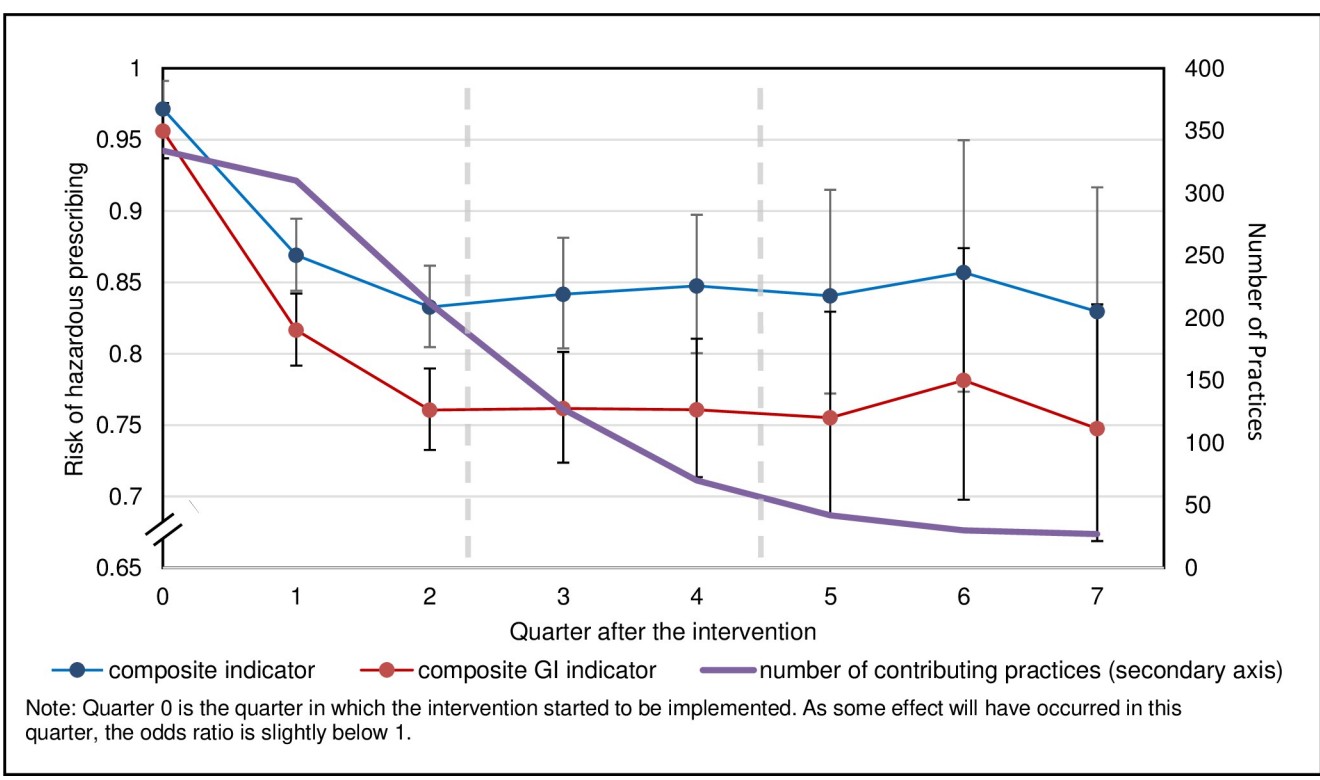

**Fig 2. Odds ratio and 95% confidence intervals for the rate of hazardous prescribing at each quarter interval compared to preintervention for the composite outcomes.**

there was (at least in the short term) a reduction in prescribing NSAIDs to patients with heart failure, and in the SMASH and DQIP studies, there was a reduction in prescribing NSAIDs to patients with chronic kidney disease. Apart from the original PINCER trial, neither the current study nor the findings from the SMASH evaluation showed that these interventions were effective at reducing the prescription of beta-blockers to patients with asthma. Anecdotal evidence suggests that this indicator is often challenging due to the indications for beta-blockers for conditions such as anxiety, and, therefore, many clinicians may opt to continue to prescribe the beta-blocker despite the risks.

The level of uptake of the intervention was very high with 94% (370/393) of eligible practices in the East Midlands of England participating in the implementation. The 370 study practices that implemented PINCER had similar characteristics to English practices as a whole, although there were some differences whereby study practices had a slightly higher list size and a higher proportion with an "outstanding" rating for safety by the CQC. Although the practices included in the study are limited to one region of England, given similar findings from the SMASH study [9] in Greater Manchester, it is likely that the intervention would be effective in other English practices.

Originally, we had planned to roll the intervention out using a randomised stepped-wedge design. However, it became clear that CCGs needed to include the rollout of PINCER in their work plans (which are set prior to each NHS financial year), and in some cases, the Medicines Optimisation Teams had to submit a business case to their CCG to enable them to repurpose their pharmacist resource. This made the process of randomisation impossible. We did, however, take account of secular trends by including the calendar reference date in the models.

This was clearly important given the trend towards safer prescribing over time (see Fig 1). The model used was an extension of the conventional segmented regression, which allows explicitly for the intervention effect to be nonlinear. We expected a priori when we specified the analysis that the treatment effect would reduce over time and wanted a model that would capture the time trends here. We were fortunate in having a very large sample size to allow a rich representation (via a simple qualitative postintervention time parameter) of these trends. Nevertheless, it should be recognised that this was an observational study, and the findings may have been influenced by unknown confounding factors or behavioural changes unrelated to the PINCER intervention.

We were not able to collect 6- and 12-month follow-up data on all practices, due to incomplete transfer of the retrospective data in some instances, and this may have biased the findings. However, a comparison between the rates for the raw data preimplementation for the full data set and separately for those practices contributing and not contributing data at 6 and 12 months postintervention (S6 Appendix) showed similar rates.

We constructed our indicators so that the denominators contained "at-risk" groups of patients but acknowledge that, in some cases, taking an action such as adding a PPI removes a patient from both the numerator and denominator for subsequent data collections. Given that the denominator is always larger than the numerator, this still means that reductions in both leads to reduction in the proportion of patients exposed to hazardous prescribing. Nevertheless, a larger effect might have been demonstrated if we had put the term "without coprescription of an ulcer-healing drug" in the numerator.

The findings may underrepresent the effectiveness of the intervention because we defined the time that practices downloaded the computer searches to identify patients at risk of prescribing as the time that each practice started the intervention. We are, however, aware that some general practices downloaded the computer searches but, due to competing priorities, did not start the intervention for several months, which would cause a delay in the effect of the intervention.

In the interrupted time series model, we have assumed that there is no residual autocorrelation. This assumption is implicit and unfortunately hard to test within the modelling framework/software we have used. A priori we expected that autocorrelation within practices would not be a substantive issue as the events of concern are intrinsically independent. We did anticipate strong correlations within practices and accounted for these in the model specification and also by utilising robust SE estimates.

Since 2015, the PINCER intervention [5] has been incorporated into national guidelines to support medicines optimisation by the National Institute for Health and Care Excellence (NICE) [11], meaning that general practices throughout the country are encouraged to use the intervention. Since this study was undertaken, several further policy developments have occurred that are helpful when discussing the place of PINCER with general practices and CCGs in England.

In 2017, PRIMIS was funded by the Health Foundation to work with Spring Impact to implement their systematic five-stage process to design a replication model for the scale and spread of PINCER using a social franchising approach [12]. As a result of this work, in 2018, PINCER was selected by the Academic Health Science Network (AHSN) for national adoption whereby PRIMIS acted as "franchisor" and the 15 AHSNs in England acted as "franchisees." Since 2018, PRIMIS has worked with all 15 AHSNs in England to roll out PINCER to over 2,800 (41%) of GP practices in England and have trained more than 2,250 healthcare professionals (including 1,713 primary care pharmacists) to deliver the PINCER intervention through a combination of eLearning tools, online resources, live webinars, and face-to-face action learning set sessions [13].

In terms of longer-term sustainability of the PINCER intervention, in 2019, NHS England set out in its long-term plan a commitment for pharmacists to take on an expanded role at the heart of local Primary Care Networks (PCNs) across the country. The new General Medical Services (GMS) contract set out the ambition for every PCN to have access to a pharmacist, thus ensuring a commitment to establishing and expanding the pharmacy workforce capable of carrying out the PINCER intervention in collaboration with CCG teams. Also, in 2019, NHS Improvement published the NHS Patient Safety Strategy, highlighting PINCER as a successful example of delivering improvement [14]. Pharmacists in general practice in England undertake a range of activities aimed at medicines optimisation, and, while delivery of the PINCER intervention is not necessarily an explicit part of the job description, there is ongoing policy drive towards the use of this approach. For example, in 2022–23, all general practices in England have been incentivised to use the PINCER approach with pharmacists undertaking structured medication reviews for patients identified from PINCER indicators [15].

Important policy questions remain in relation to the effectiveness of the intervention at translating reductions in hazardous prescribing to reducing serious harm to patients, and the cost-effectiveness of the intervention. The DQIP [8] intervention was associated with a reduction in hospital admissions for GI bleeding and heart failure, and our detailed economic evaluation of the original PINCER intervention suggested it produced marginal health gain at slight reduced overall cost [16]. These findings are reassuring in terms of the likely benefits of rolling out PINCER at scale, but in order to give a definitive answer to these questions, we are currently evaluating the effectiveness of the national rollout of the PINCER intervention in reducing serious patient harm, particularly from GI bleeding as part of our NIHR Programme Grant for Applied Research "Avoiding patient harm through the application of prescribing safety indicators in English general practices (PRoTeCT)"[17].

In conclusion, the PINCER intervention, when rolled out at scale in routine clinical practice, was associated with a reduction in hazardous prescribing by 17% and 15% at 6 and 12 months postintervention. The greatest reductions in hazardous prescribing were for indicators associated with risk of GI bleeding, particularly where prescription of an ulcer-healing drug would improve patient safety.

## Supporting information

**S1 Appendix. Rollout of the PINCER intervention across general practices in 11 CCGs.**
(PDF)

**S2 Appendix. The Explicit Statistical Model.**
(PDF)

**S3 Appendix. STROBE Checklist.**
(PDF)

**S4 Appendix. The number of practices included at each quarter, by time since the intervention.**
(PDF)

**S5 Appendix. Comparing the rate of hazardous prescribing at 6 months and 12 months postintervention to preintervention, adjusted for GP practice and calendar time (excluding practices with less than 6 months of data postintervention).**
(PDF)

**S6 Appendix. Comparing the baseline (preimplementation year) rates of hazardous prescribing for practice with at least and without at least 6 months and 12 months**

**postintervention data: Raw data and data adjusted for GP practice and calendar time.**
(PDF)

**S7 Appendix. Plots for each Individual Indicator.**
(PDF)

## Acknowledgments

We thank:

- Members of the Project Steering and Evaluation Advisory Committees who provided extremely valuable advice and support throughout the course of the project.

- The Clinical Commissioning Groups (CCG), medicines management teams and general practices that participated.

- The pharmacists and pharmacy technicians who delivered the PINCER intervention.

- CCG research and development leads, pharmacy leads, prescribing advisors and other key individuals who helped to facilitate the project including Rachel Illingworth, Harriet Murch, Mindy Bassi and David Gerrett.

- Tony Panayiotidis and Lauren Fensome, PRIMIS, University of Nottingham for delivering training on CHART software.

- PRIMIS colleagues for collection of retrospective data and support throughout the project

- Gill Gookey for delivering training on the PINCER intervention.

- The following members of the PINCER implementation team: Mindy Bassi, Dr Cheryl Crocker, Despina Laparidou, Tony Panayiotidis.

- Lincolnshire Community Health Services NHS Trust for hosting the grant and for oversite of the project finances.

- East Midlands Patient and Public Involvement senate for their valuable advice and ongoing help and support.

- Antony Chuter, Chris Rye and Glen Swanwick for providing Patient and Public Involvement and Engagement.

- Christina Sheehan for oversite of the evaluation finances.

- Sarah Armstrong and Rajnikant Mehta, NIHR RDS for the East Midlands, University of Nottingham, for advice on statistical analysis.

## Author Contributions

**Conceptualization:** Sarah Rodgers, Darren M. Ashcroft, James Barrett, Matthew J. Boyd, Rachel A. Elliott, Kamlesh Khunti, Aziz Sheikh, Aloysius Niroshan Siriwardena, Anthony J. Avery.

**Data curation:** Sarah Rodgers, Amelia C. Taylor.

**Formal analysis:** Amelia C. Taylor, Stephen A. Roberts.

**Funding acquisition:** Sarah Rodgers, Darren M. Ashcroft, Matthew J. Boyd, Rachel A. Elliott, Kamlesh Khunti, Aziz Sheikh, Aloysius Niroshan Siriwardena, Anthony J. Avery.

**Investigation:** Sarah Rodgers, James Barrett, Despina Laparidou, Aloysius Niroshan Siriwardena, Anthony J. Avery.

**Methodology:** Sarah Rodgers, Amelia C. Taylor, Stephen A. Roberts, Thomas Allen, Darren M. Ashcroft, Rachel A. Elliott, Anthony J. Avery.

**Project administration:** Sarah Rodgers, Amelia C. Taylor, Anthony J. Avery.

**Resources:** Sarah Rodgers, James Barrett, Anthony J. Avery.

**Software:** Amelia C. Taylor, Stephen A. Roberts.

**Supervision:** Sarah Rodgers, Anthony J. Avery.

**Validation:** Sarah Rodgers, Amelia C. Taylor, Stephen A. Roberts.

**Visualization:** Sarah Rodgers, Amelia C. Taylor, Stephen A. Roberts, Anthony J. Avery.

**Writing – original draft:** Sarah Rodgers, Amelia C. Taylor, Stephen A. Roberts, Anthony J. Avery.

**Writing – review & editing:** Sarah Rodgers, Amelia C. Taylor, Stephen A. Roberts, Thomas Allen, Darren M. Ashcroft, James Barrett, Matthew J. Boyd, Rachel A. Elliott, Kamlesh Khunti, Aziz Sheikh, Despina Laparidou, Aloysius Niroshan Siriwardena, Anthony J. Avery.

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
