## [Editor Report · Decision Letter 0]

21 Mar 2022

Dear Dr Taylor, 

Thank you for submitting your manuscript entitled "Scaling-up a pharmacist-led information technology intervention (PINCER) to reduce hazardous prescribing in general practices: multiple interrupted time series study" for consideration by PLOS Medicine.

Your manuscript has now been evaluated by the PLOS Medicine editorial staff and I am writing to let you know that we would like to send your submission out for external peer review.

Please re-submit your manuscript within two working days, i.e. by Mar 23 2022 11:59PM.

Kind regards,

Beryne Odeny

PLOS Medicine

---

## [Decision Letter · Decision Letter 1]

1 Jun 2022

Dear Dr. Taylor,

Thank you very much for submitting your manuscript "Scaling-up a pharmacist-led information technology intervention (PINCER) to reduce hazardous prescribing in general practices: multiple interrupted time series study" (PMEDICINE-D-22-00818R1) for consideration at PLOS Medicine. 

Your paper was evaluated by a senior editor and discussed among all the editors here. It was also sent to independent reviewers, including a statistical reviewer. The reviews are appended at the bottom of this email and any accompanying reviewer attachments can be seen via the link below:

[LINK]

In light of these reviews, I am afraid that we will not be able to accept the manuscript for publication in the journal in its current form, but we would like to consider a revised version that addresses the reviewers' and editors' comments. Obviously we cannot make any decision about publication until we have seen the revised manuscript and your response, and we plan to seek re-review by one or more of the reviewers. 

We expect to receive your revised manuscript by Jun 22 2022 11:59PM. Please email us (plosmedicine@plos.org) if you have any questions or concerns.

We look forward to receiving your revised manuscript. 

Sincerely,

Beryne Odeny, 

PLOS Medicine

plosmedicine.org

1) The Data Availability Statement (DAS) requires revision. For each data source used in your study:

2) Please remove the ‘Funding,” “competing interests” and “Data availability statement” after the title page. In the event of publication, this information will be published as metadata based on your responses to the submission form.

3) Abstract:

a) Please structure your abstract using the PLOS Medicine headings (Background, Methods and Findings, Conclusions). Please replace the subheading “Objectives” with “Background”

b) Please ensure that all numbers presented in the abstract are present and identical to numbers presented in the main manuscript text.

c) Please quantify the main results (with 95% CIs and p values).

d) Please provide the actual numbers of events for the outcomes (numerator and denominator), not just summary statistics or ORs.

e) In the last sentence of the Abstract Methods and Findings section, please describe the main limitation(s) of the study's methodology.

4) Author summary - At this stage, we ask that you reformat your non-technical Author Summary. The Author Summary should immediately follow the Abstract in your revised manuscript. This text is subject to editorial change and should be distinct from the scientific abstract. The summary should be accessible to a wide audience that includes both scientists and non-scientists. Please see our author guidelines for more information: https://journals.plos.org/plosmedicine/s/revising-your-manuscript#loc-author-summary.

5) Your study is observational and therefore causality cannot be inferred. In the abstract and discussion, please remove language that implies causality, such as “reduced hazardous prescribing” or similar. Refer to associations instead.

6) Please add the following statement, or similar, to the Methods: "This study is reported as per the Strengthening the Reporting of Observational Studies in Epidemiology (STROBE) guideline (S1 Checklist)."

7) When completing the STROBE checklist, please use section and paragraph numbers, rather than page numbers.

8) In the methods and results please address the following:

a) In the main text, please provide the actual numbers (numerator and denominator) for the outcomes, not just summary statistics or ORs.

b) Please present both numerators and denominators for rates, in the Table 4 

c) Please provide both adjusted analyses and unadjusted analyses, where appropriate

d) In the text, please provide p values in addition to 95% CIs when describing numeral results

e) When a p value is given, please specify the statistical test used to determine it.

9) In Figure 2 and S4, please show the Y axis beginning at zero. If this is not possible, please show a break in the axis.

10) In the tables, please define all abbreviations e.g., CCG, Q, GI

11) In the table S3 footnotes, please define what Indicator A, B, C… stand for.

12) Please remove subheadings in the Discussion section, and present and organize the Discussion as follows: a short, clear summary of the article's findings; what the study adds to existing research and where and why the results may differ from previous research; strengths and limitations of the study; implications and next steps for research, clinical practice, and/or public policy; one-paragraph conclusion.

13) References:

a) Please ensure that journal name abbreviations consistently match those found in the National Center for Biotechnology Information (NCBI) databases. https://journals.plos.org/plosmedicine/s/submission-guidelines#loc-references. 

b) Please include access dates for all weblinks and ensure that all weblinks are current and accessible, e.g. ref #4

Comments from the reviewers:

Reviewer #1: 1. Summary of the research and your overall impression

This is an observational study of the implementation of a pharmacist-led information technology intervention for medication errors in 343 GP practices between September 2015 and August 2016. 

Electronic health care records were analysed to identify pre-specified patients at risk of harm from prescribed treatments. This is was same for the intervention and the analysis. 

This intervention had previously been studied as a multicentre, cluster randomised, controlled trial in 72 GP practices in 2006; 32 in the intervention arm. Other similar studies were implemented in 2011-12 (DQIP) and 2017-17 (SMASH). 

The authors reported findings are a reduction in the number of patients with hazardous prescribing. This reduction is limited to hazardous prescribing associated with gastro-intestinal bleeding (5 out of 6 individual measures), individual measures for exacerbation of asthma, heart failure stoke kidney injury had no observed reduction. 

It was pleasing to see patient and public involvement.

2. Discussion of specific areas for improvement

Issue 1

The authors state in line 143 that data was extracted retrospectively. That up to 16 quarters were extracted (Dec 2013 to Aug 2017). Then in 147 they state that not all practice contributed 16 quarters of data and that for most data was extracted in Oct or Nov 2017. 

The number of practices included in the analysis changes with time and this is summarised in table S2. The quarter before the intervention period (quarter -1) there are 341 practices included and for the 7 quarters preceding this there are between 341 and 343 practices. For the period of time at the start of the intervention there were 343 practices, then 310 in the following quarter (quarter +1) and 212 in the next (quarter +2). By quarter +4 there were 70 practices and by quarter +7 there are only 27 practices. 

This means the primary statistical analysis of improvement at 6 months (quarters +2) was based on approximately two thirds of practices and the secondary analysis at 12 months (quarters +4) in one fifth of practices. 

It could be argued that the paper is redrafted to only include the 212 practices with sufficient data to analyse the primary end point with a sensitivity analysis conducted with the subset of practice (70) with sufficient data to analyse the secondary end point. The other 131 practices (343 less 212) are contributing data to the baseline analysis but not the post-intervention period and estimation of intervention effect size. 

As a minimum the number of GP practices contributing data to the primary and secondary end points should be made clearer at the start of the results section (currently it is at the end - line 284). Adding the number of practices contributing data to each quarter to the Figure 2 and S4 plots would help make this clearer.

Issue 2

The analysis method is described in the title as an interrupted time series analysis (ITSA). 

The statistical methodology estimates a single pre-implementation base rate (mean of the 4 quarters immediately pre-intervention) with post-intervention values estimated at 6 months and 12 months from temporal trends. However, this appears to mean that pre-intervention trends are not used to estimate the effect size from the counterfactual trend line and that the temporal trends estimate in the post intervention period for the primary outcome are based on two values (quarter +1 and quarter +2). The minimum number recommended for Cochrane systematic reviews of ITSA is three pre and three post intervention. It would appear that according to Cochrane criteria this study would not be included in an Effective Practice and Organisation of Care (EPOC) review https://epoc.cochrane.org/sites/epoc.cochrane.org/files/public/uploads/inttime.pdf

A review of the statistical methodology by a statistician would be useful to assess the appropriateness of the statistical method and the use of the description "interrupted time series study".

Other minor issues

1. The table footnotes in Table S3 are missing (a,b,c,d are not defined)

2. CIs in Table S3 12months OR (last column) have additional spaces which do not exist for the other CIs in Table 4 or Table S3

Reviewer #2: This large study explored the effectiveness of the PINCER intervention (a pharmacist-led information technology intervention for medication errors ) to identify patients at risk from hazardous prescribing and to reduce this risk across over 90% of GP practices in one region of England. The original PINCER study found that the outcome in terms of hard reduction, was greater with pharmacist follow up that with computerised feedback alone. This paper answers an important question; are the findings from the PINCER study transferable in real-life setting. However, whilst the authors confirm that indeed this approach can and does reduce the rates of hazardous prescribing there are a few questions which need to be addressed before it can be accepted for publication. Firstly, whilst almost 350 practices were recruited, data from only 212 were analysed at 6 months and only 70 at 12 months. The authors have included a sensitivity analysis to determine whether including data from practices where they have full data at 6 months produces the same outcomes as the partial data analysis and have confirmed that this is the case. However, they have not provided any explanation as to why the drop out rate is so high nor have they considered that those practices with full-data at 6 months may be more engaged and therefore have improved outcomes compared to those who dropped out? What are the differences between the characteristics of the practices that data was obtained at 6 and 12 months compared to those that started the initiative? Was the difference in engagement due to level of pharmacist support between practices? No information is provided in the manuscript regarding this detail. Particularly given that the PINCER study demonstrated better results with pharmacist support this is important. Figure 2 shows that the 95% confidence intervals post intervention (for GI and all outcomes) are much greater than pre-intervention. Why is this? Does this suggest that the intervention becomes less effective over time for some practices? Why? - is the presence or otherwise of a pharmacist contributing to this variability or are those practices who are rated as poor/ needs improvement by the CQC less likely to be following up on the recommendations with time or is there another explanation? The authors also fail to comment on why the prescribing of long acting beta-2 agonists to patients with asthma who were not receiving inhaled steroids was so high nor why it failed to improve at all with the intervention. Please clarify why outcome J is labelled stroke - the prescription of antipsychotics to patients with dementia without a diagnostic code of psychosis. 

Reviewer #3: Thank you for submitting this interesting study. I have a few comments and questions for the authors to consider and address.

Line 118: I appreciate these indicators are from the original PINCER trial - but could you provide a reference to how these indicators were informed.

Line 123: Please provide more information regarding the pharmacists involved in the intervention. Were they co-located in the general practice? How many hours per week were they working in the general practice? What experience/training did they have prior to the study? Were they experienced general practice pharmacists? Were they prescribers?

Line 126: How often did these discussions take place, where and when did they take place?

Line 127: Please provide an example of these educational materials.

Line 128: Was the action plan retained in the general practice?

Line 130-131: This section states that pharmacists supported by technicians implemented the action plan. Were these pharmacists/technicians in the community setting? If so, where was the action plan retained? How did the pharmacist actually "support general practice staff" in implementing the action plan?

Line 146: Although the authors have referenced the original paper, it would be good to have an idea of what this processing included.

Line 156-157: "For most practices, retrospective data extraction was carried out during October 2017 and November 2017." Does this mean that practices only carried out data extraction at two time points?

Line 149-198: My main comment about this section is that it is quite hard to follow when data were collected, extracted, action plans were implemented etc. Would suggest revising how this is reported and perhaps include a timeline or the similar.

Line 166: Why were the gastro-intestinal indicators singled out as a group?

Line 287: "We had complete data for all 343 practices." What does this mean? Does it mean that data were collected for all practices at all quarters? This links back to my comment that the timeline of data collection and analysis is difficult to follow.

Line 314: "The level of uptake of the intervention was very high with 94% (370/393) of eligible practices in the East Midlands of England participating in the implementation." In order to be counted as having participated, did the practice only have to submit one set of data at one quarter, i.e. they didn't have to continuously submit data at all quarters?

Line 322: What were these pragmatic reasons?

Line 323: What were these "secular trends"? 

Line 389-393: Although HCPs have been trained in PINCER, do the authors have any idea as to how much it is actually used in day-to-day practice?

Line 396-399: Can the authors better explain the relationship between pharmacists being employed in general practice and increased use of PINCER - is use of PINCER an explicit part of their job description?

Reviewer #5: The study uses an interrupted time series approach to evaluate the rate of hazardous prescribing after implementing the PINCER intervention. I found this to be an interesting and generally well presented study. My comments mainly concern the detail given on the statistical model used and its assumptions.

- Although the statistical model is described in words, I think it would be beneficial to also provide the model explicitly, perhaps in the supplementary material. This is not a straightforward model and would be clearer written out in equation form with full details given.

- Is it assumed that there is no time-dependant trend prior to intervention? The definition of base rates as an average of four quarters prior to implementation suggests it may be, although, again, this would be clearer if the model was written out. Is this a reasonable assumption? Could the data be plotted where the x-axis is 'quarter relative to intervention' (possibly after accounting for secular trend)?

- Another assumption of the interrupted time series model is that there is no residual autocorrelation. Is this justifiable? Could this be commented on and/or tested?

- Is it possible that a patient was at risk of or exposed to more than one hazardous prescribing indicator? The descriptions in Table 1 and the data in Table 3 suggests that this may be the case. Has any overlap been accounted for when calculating the numerators and denominators of the composite indices?

[LINK]

---

## [Decision Letter · Decision Letter 2]

26 Sep 2022

Dear Dr. Taylor,

Thank you very much for re-submitting your manuscript "Scaling-up a pharmacist-led information technology intervention (PINCER) to reduce hazardous prescribing in general practices: multiple interrupted time series study" (PMEDICINE-D-22-00818R2) for review by PLOS Medicine.

I have discussed the paper with my colleagues and it was also seen again by three reviewers. I am pleased to say that provided the remaining editorial and production issues are dealt with we are planning to accept the paper for publication in the journal.

[LINK]

We look forward to receiving the revised manuscript by Oct 03 2022 11:59PM.   

Sincerely,

Beryne Odeny, 

PLOS Medicine

plosmedicine.org

Requests from Editors:

1) Abstract: 

a) In the last sentence of the Abstract “Methods and Findings section,” please describe the main limitation(s) of the study's methodology. You indicate that this has been added; please ensure that you have situated in the right place within the abstract. It looks like you added the statement to the Abstract’s Conclusions – please move it to the “Methods and findings” instead.

b) Please interpret the study based on the results presented in the abstract, emphasizing what is new without overstating your conclusions. In addition please state specific implications of your study

2) Author summary:

a) Please put the limitations statement under the subheading “What Did the Researchers Do and Find?”

b) Author summary - Please avoid vague statements such as "these results have major implications for policy/clinical care". Mention only specific implications substantiated by the results.

3) Thank you for providing a completed STROBE checklist. Please remove the Column, Line No, as this will likely change in the event of publication. Instead provide the paragraph number in addition to the section number.

Comments from Reviewers:

Reviewer #4: Dear editors,

I would like to express my compliments to the authors for their careful revision, detailed responses and additional analyses. My concerns regarding the statistical analyses and specification of indicators have been addressed. My concerns regarding loss to follow up have only been partially addressed but the authors have done what is in their power to address them. I therefore recommend publication combined with a request for a minor correction. In response to comment 54, the authors now state that: "Nevertheless, a larger effect might have been demonstrated if had put the term "without co-prescription of an ulcer healing drug in the denominator". The word denominator should be replaced by the word numerator.

Reviewer #5: I thank the authors for their responses to my comments, which have been addressed satisfactorily.

[LINK]

---

## [Editor Report · Decision Letter 3]

21 Oct 2022

Dear Dr Taylor, 

On behalf of my colleagues and the Academic Editor, Professor Aaron Kesselheim, I am pleased to inform you that we have agreed to publish your manuscript "Scaling-up a pharmacist-led information technology intervention (PINCER) to reduce hazardous prescribing in general practices: multiple interrupted time series study" (PMEDICINE-D-22-00818R3) in PLOS Medicine.

Before your manuscript can be published you will need to address the final revision detailed below:

AUTHOR SUMMARY – What do these findings mean?

Line 95: “This has important implications for policy makers considering the roll out of similar interventions.” Please revise this section to read as follows: 

What Do These Findings Mean? 

• The findings suggest that the PINCER intervention was effective when rolled out at scale 

• These findings support the wider national rollout of PINCER in England and may help to inform policy makers when considering implementation of similar interventions.

PRESS

Sincerely, 

Pippa

Philippa Dodd, MBBS MRCP PhD 

Editor 

PLOS Medicine